# Myocardial Segmentation of Tagged Magnetic Resonance Images with Transfer Learning Using Generative Cine-To-Tagged Dataset Transformation

**DOI:** 10.3390/bioengineering10020166

**Published:** 2023-01-28

**Authors:** Arnaud P. Dhaene, Michael Loecher, Alexander J. Wilson, Daniel B. Ennis

**Affiliations:** 1Department of Radiology, Stanford University, Stanford, CA 94305, USA; 2Signal Processing Laboratory (LTS4), École Polytechnique Fédérale de Lausanne (EPFL), 1015 Lausanne, Switzerland; 3Stanford Cardiovascular Institute, Stanford University, Stanford, CA 94305, USA

**Keywords:** myocardial segmentation, deep learning, MRI tagging, cardiac MRI, domain transformation, dataset augmentation

## Abstract

The use of deep learning (DL) segmentation in cardiac MRI has the potential to streamline the radiology workflow, particularly for the measurement of myocardial strain. Recent efforts in DL motion tracking models have drastically reduced the time needed to measure the heart’s displacement field and the subsequent myocardial strain estimation. However, the selection of initial myocardial reference points is not automated and still requires manual input from domain experts. Segmentation of the myocardium is a key step for initializing reference points. While high-performing myocardial segmentation models exist for *cine* images, this is not the case for tagged images. In this work, we developed and compared two novel DL models (nnU-net and Segmentation ResNet VAE) for the segmentation of myocardium from tagged CMR images. We implemented two methods to transform cardiac *cine* images into tagged images, allowing us to leverage large public annotated *cine* datasets. The cine-to-tagged methods included (i) a novel physics-driven transformation model, and (ii) a generative adversarial network (GAN) style transfer model. We show that pretrained models perform better (+2.8 Dice coefficient percentage points) and converge faster (6×) than models trained from scratch. The best-performing method relies on a pretraining with an unpaired, unlabeled, and structure-preserving generative model trained to transform *cine* images into their tagged-appearing equivalents. Our state-of-the-art myocardium segmentation network reached a Dice coefficient of 0.828 and 95th percentile Hausdorff distance of 4.745 mm on a held-out test set. This performance is comparable to existing state-of-the-art segmentation networks for *cine* images.

## 1. Introduction

Cardiovascular disease is the leading cause of death globally. One key metric for assessing cardiac performance is ejection fraction (EF), which is measured using a type of magnetic resonance imaging (MRI) sequence termed *cine* (i.e., dynamic) imaging. A different MRI sequence termed, tagged MRI, is the clinical reference standard for measuring myocardial strain [1], which can be measured along the radial, circumferential and longitudinal directions (Figure 1a and Figure 2). Myocardial strain has a high prognostic value for diagnosing cardiovascular diseases [2]. For example, pathologies such as concentric hypertrophy and eccentric hypertrophy may exhibit the same EF, but can be distinguished by their radial and circumferential strain measures [3].

Tagged CMR imaging is not currently included in a standard clinical workflow due to time-consuming analysis and the need for specialized software [1]. Recently there has been substantial progress towards automating this processing, such as the development of models that automate strain analysis of tagged CMR image sets [6,7]. However, the model requires manual input of myocardial reference points by a domain expert, and thus the initialization of reference points is currently a barrier to full automation of this process.

In recent years, the most widely used methods in medical image segmentation have been based on DL. Within cardiac MRI, the most common segmentation task is delimiting cardiovascular anatomical structures such as the chambers and the coronary arteries. An example from the Automated Cardiac Diagnosis Challenge (ACDC) is presented in Figure 1b. The ACDC also provided a public dataset of cardiac *cine* images and annotations. The quantification of these structures allows for the measurement of chamber volumes, myocardial mass, wall thickness, and EF. State-of-the-art models for the segmentation of the myocardium and the ventricular chambers from cardiac *cine* are typically DL approaches using Unet and ResNet architectures [8,9]. Herein, we adapt these types of architectures for the development of a myocardial segmentation tool for tagged MRI. The state-of-the-art *cine* segmentation models are able to achieve a dice coefficient of 0.849 [10]. This provides a performance reference for the current work.

However, the training of a segmentation network for cardiac tagged images requires a large data set of tagged images. Currently, there is a paucity of publicly available tagged CMR data sets with myocardial annotations. We have identified this as a key obstacle to the development of a tagged segmentation model. However, there is a relative abundance of publicly available cardiac cine images with annotations. Herein, we make use of this data for training the tagged segmentation model by transforming the *cine* images into pseudo-tagged images.

The transformation of cine to pseudo-tagged images can be framed as a style transfer problem. Within the DL literature, there has been recent development of models that are able to accurately learn an image "style" and then apply a transformation to an image of a different style. Cyclic Generative Adversarial Networks are neural network architectures that are well suited to this problem [11]. In recent years, GAN-based models have gained a lot of traction within the field of medical image computing due to their conditional image generation ability. They have achieved impressive results in a range of tasks including image editing [12], representation learning [13], and image segmentation [14]. We make use of a CycleGAN approach to develop a *cine*-to-tagged style transfer model. As a comparison, we also use an algorithmic approach based on MRI physics. We investigate the importance of this pseudo-tagged data for training the segmentation models, particularly the time-to-convergence and overall performance as measured by the Dice coefficient and Hausdorff distance.

The objective of this work was to develop a myocardial segmentation neural network for tagged CMR images in the absence of a large database of annotated tagged images for training. To do so, we utilized large public *cine* image datasets with annotations and transformed these into their tagged-appearing equivalents for segmentation model pretraining. Herein, we demonstrate an unpaired, unlabeled, and generative *cine*-to-tagged image transformation method that leverages the existence of publicly available *cine* data and mitigates the scarcity of clinical resources for manual annotation of tagged images. Finally, we developed a state-of-the-art segmentation model for detecting the LV myocardium in tagged images.

## 2. Materials and Methods

### 2.1. Unpaired and Unlabeled *Cine* to Tagged Image Transformation

A physics-driven transformation and a DL generative model were developed to transform *cine* images into their tagged-appearing equivalents. The related methods and implementation choices are presented in this section.

#### 2.1.1. Public Annotated *Cine* Datasets

While public datasets of CMR images exist, the overwhelming majority are dynamic images acquired with a bSSFP sequence, also referred to as *cine* images (Figure 2, top). Despite the identical underlying anatomical structure, the image contrast and signal features are different in tagged images, making it more difficult to discern and identify cardiovascular structures. Herein we develop two distinct data transformation techniques that leverage (1) acquisition physics and (2) unlabeled unpaired data to transform *cine* images into tagged images. Using these models, large public datasets of annotated *cine* images (Table 1) are transformed into their tagged-appearing equivalents.

#### 2.1.2. Physics-Driven *Cine* to Tagged Data Transformation

A physics-driven transformation is implemented consisting of modifying the image contrast and adding tag lines to the image. Contrast modification is performed by taking the element-wise square root of the image, which grossly approximates a bSSFP to gradient echo (GRE) image contrast transformation. Tag lines are added on an equally spaced rotated grid using a physics-based approach where the Bloch equations are used to simulate the RF and gradient events used in a standard tagging sequence. The tagging preparation used a 1-3-3-1 binomially weighted RF pulse train. Contrast dynamics, line tag spacings and flip angles were all designed to have realistic contrast and were derived from parameters determined in [16].

The transformation was implemented in Python and is dependent on two parameters, the tag-line spacing and the RF flip of the tagging pulse. In our implementation, we use 5-pixel spacing between tag lines and a total tagging flip angle of 70 degrees, which was selected empirically to mimic somewhat faded tag lines. A notable limitation of the transformation is that tag lines do not deform according to the acquisition time frame.

#### 2.1.3. Deep Learning Based *Cine* to Tagged Data Transformation

An unpaired image-to-image transformation translation neural network (i.e., CycleGAN) was trained using an unlabeled and unpaired collection of *cine* and tagged images. The model consists of four sub-networks, two generators and two discriminators. We modeled both the transformation from *cine* to tagged and vice versa are modeled (Figure 3). This architecture was chosen for its cycle-consistent optimization loss, ensuring that transformed images are optimized not to displace inherent cardiac structures [11]. In our work, the myocardium segmentation mask was transferred from its original *cine* image to the newly generated tagged equivalent. A CycleGAN as described in the original paper by Zhu et al. (2020) was trained for 200 epochs on the aforementioned dataset containing 2440 *cine* and 4539 unlabeled tagged images. As the model is trained on images through acquisition time (full cycle, from ED to ES to ED), it is theoretically able to deform taglines according to the myocardial contraction found in the images from the tagged domain.

### 2.2. Segmentation of the Myocardium in Tagged CMR Images

Labeled tagged images are scarce due to clinical resource constraints and time-intensive annotation procedures. The approach taken in this work for model development relies heavily on transfer learning. The main concept revolves around pre-training common architectures with simulated tagged images, transformed by the models detailed in Section 2.1. Then, models were fine-tuned on a small subset of manually annotated data.

#### 2.2.1. Dataset

The dataset of real tagged images is comprised of scans from fifty pediatric patients that are either healthy control subjects (n = 20) or patients diagnosed with Duchenne’s Muscular Dystrophy (DMD, n = 30). This dataset was chosen due to its availability and relevance to diagnosing pediatric heart disease. Each sequence of images contains twenty-five time-resolved 2D slices. The dataset was split into 1000 2D images for training purposes and 250 for testing purposes. The test set was balanced in terms of DMD patients and healthy subjects. The myocardium was annotated in each image by an expert from the Radiological Sciences Laboratory at Stanford University.

#### 2.2.2. Training Strategy

The ACDC, M&Ms, and SCD datasets (Table 1) are transformed with either the physics-driven transformation algorithm or the generative model $G_{A\to B}$ to yield training datasets of 4578 tagged-appearing images. The annotated regions of interest were transferred from the original datasets for each transformed image. Using these transformed datasets, four different training strategies were compared:Train from scratch.Pretrain with public datasets of *cine* images.Pretrain with public datasets of *cine* images, transformed into a simulated tagged domain with the physics-driven method.Pretrain with public datasets of *cine* images, transformed into a simulated tagged domain with the $G_{A\to B}$.

The models that were not pretrained (i.e., training strategy 1 as well as the pretraining runs themselves) undergo Xavier normal weight initialization [17].

#### 2.2.3. Network Architectures

We compare the performance of two distinct network architectures widely used in medical segmentation tasks; nnUnet [8] and a segmentation Variational Auto-Encoder (VAE) based on residual networks (ResNet) [9], which we will call ResNetVAE. We used the open-source implementation of these network architectures found in the MONAI Python library [18]. Both of these architectures are based on an underlying encoder-decoder pipeline. First, the image is encoded in a latent space representation of lower dimensionality than the input. Then, this representation is decoded into a pixel-wise segmentation map.

There are two main differences between the nnUnet and the ResNetVAE architectures. First, the nnUnet is larger, requiring approximately ten times more parameters to optimize. The nnUnet has 13 blocks (one input block, 5 downsample blocks, one bottleneck and 6 upsample blocks) and an according 26 activation functions compared to the ResNetVAE’s 7 blocks and 16 activation functions. While additional capacity can increase performance, it can also lead to overfitting of the training data. The second main difference is that the latent space of the ResNetVAE is regularized as a normal distribution. This regularization leads to attractive properties in the decoding process as it constrains features in the latent space to have a low degree of inter-correlation. The initial image processing methodology consists of two steps: (1) resizing the image to 256 × 256 (not necessary for the nnUnet) and (2) normalization of the input image using a mean of 0.456 and a standard deviation of 0.224.

#### 2.2.4. Train-Time Data Augmentation

The training was performed with train-time augmentations. More specifically, a stochastic combination of random flips, additive random noise and blurring, and both affine and elastic transformations is performed sequentially at every epoch and for each batch.

#### 2.2.5. Optimization

A gradient scaler was used for training in parallel on two TITAN RTX 24GB GPUs while a scheduler reduces the learning rate (initially set to 0.01) with logarithmic steps when validation loss hits a plateau with a patience value of ten epochs. The equally weighted sum of the Dice coefficient loss and pixel-wise cross-entropy loss was used as a loss function to optimize the network’s weights with Adam stochastic optimization [19].

Additionally, a shape distance loss was implemented to give more importance to the ground segmentation masks’ boundaries. This loss is based on the pixel-wise difference between the sigmoid of the log odds prediction values and a boundary distance map of the ground-truth label.

Let us denote *g* and *p* as the multi-channel binary ground-truth and prediction masks (each having *C* channels) and ϕ as a signed normalized Euclidean distance map operator. Shape information for a binary mask *m* is computed as follows in Equation (Equation 2), which is in turn used to compute a boundary-aware loss $L_{S}\left(g,p\right)$ defined in Equation (Equation 3) and visualized in Figure 4. The parameter *k* in Equation (Equation 1) adjusts the softness of the boundary. In this work, it is set to 0.2 based on empirical tuning.
(1)$$H\left(x\right)=\frac{1}{1+e^{-x/k}}$$
(2)$$SI\left(m\right)=\left\{\begin{array}{cc} H(1-\phi (m)) & \mathrm{if}\;\phi (m)\geq 0\\ 0 & \mathrm{otherwise} \end{array}\right.$$
(3)$$L_{S}\left(g,p\right)=\frac{1}{C}\sum_{c=1}^{C}\frac{1}{\sum g_{ci}}\int_{i\in \Omega }\left|p_{ci}-SI{\left(g_{c}\right)}_{i}\right|\;di$$

A grid search is performed to find the optimal weight contribution γ of this shape-aware loss on a ResNetVAE model trained from scratch on the manually annotated dataset of tagged images. The mathematical formulation of the loss function is presented in Equation (Equation 4).
(4)$$\begin{array}{c} L\left(g,p\right)=\underset{L_{D}\left(g,p\right)}{\underset{︸}{\left(1-\frac{1}{C}\sum_{c=1}^{C}\frac{2g_{c}p_{c}}{g_{c}+p_{c}}\right)}}-\underset{L_{CE}\left(g,p\right)}{\underset{︸}{\left(\frac{1}{C}\sum_{c=1}^{C}\int_{i\in \Omega }g_{ci}\log\left(p_{ci}\right)\;di\right)}}\\ \;\;\;\;\;\;\;\;\;\;\;\;\;\;\;\;\;+\gamma \underset{L_{S}\left(g,p\right)}{\underset{︸}{\left(\frac{1}{C}\sum_{c=1}^{C}\frac{1}{\sum_{i}g_{ci}}\int_{i\in \Omega }\left|p_{ci}-SI{\left(g_{c}\right)}_{i}\right|\;di\right)}} \end{array}$$

#### 2.2.6. Evaluation Metrics

For model evaluation, it is important to use robust and meaningful performance evaluation metrics to validate the developed algorithms [20]. In this work, we evaluate our semantic segmentation task using the Dice coefficient Equation (Equation 5) as an overlap-based metric and the 95th-percentile Hausdorff distance Equation (Equation 6) as a boundary-based metric.

Let *g* and *p* be the ground-truth and prediction masks, then $G=\sum_{i}g_{i}$ and $P=\sum_{i}p_{i}$.
(5)$$DSC\left(G,P\right)=\frac{2|G\cap P|}{\left|G|+|P\right|}$$

Let *X* and *Y* be two distinct point sets and *d* the euclidean distance between points. In the case of semantic segmentation, these point sets are considered as the contours of the ground truth and predicted segmentation masks *g* and *p*. Let *Q* be the quantile function.
(6)$$HD_{0.95}\left(X,Y\right)=\max\left(d_{0.95}\left(X,Y\right),d_{0.95}\left(Y,X\right)\right),\;\;\mathrm{with}\;\;d_{k}\left(X,Y\right)=Q_{x\in X}\left(k;\min_{y\in Y}d\left(x,y\right)\right)$$

## 3. Results

### 3.1. Unpaired and Unlabeled *Cine* to Tagged Image Transformation

Qualitatively, the physics-driven approach maintains the structure of the original *cine* image. Whilst contrast is modified, the produced images’ tagged style remains unrealistic, mostly due to contrast differences between LV, blood pool, and surrounding air in the lungs (Figure 5, center). On the other hand, the CycleGAN approach yields images that look extremely realistic in terms of contrast and style (cf. Figure 2). As in real tagged images, some anatomical structures are difficult to discern. This sometimes appears exaggerated by hallucinations. In addition, the masks transferred from the original *cine* images delineate the myocardium well in the CycleGAN tagged images, validating the hypothesis that cycle-consistent optimization successfully constrains the location of relevant physiological structures (Figure 5; bottom).

### 3.2. Segmentation of the Myocardium in Tagged CMR Images

For the DL segmentation approach, we discuss the modeling results in qualitative and quantitative manners by comparing model architectures, training strategies, and the impact of shape-aware loss.

#### 3.2.1. Model Architecture

In this work, two distinct model architectures were implemented and tested: ResNetVAE and nnUnet. Whilst they are both based on an encoder-decoder mechanism, the more commonly used nnUnet has approximately ten times more trainable parameters than ResNetVAE, as detailed in Table 2. Above that, the variational auto-encoder of ResNetVAE ensures that training is regularised to avoid overfitting by constraining the distribution of the latent space.

While all segmentations seem to present a consistent disc-looking shape, the masks produced by the ResNetVAE seem to be physiologically more consistent than the ones produces by the nnUnet (Figure 6 and Figure 7). A common prediction error is the inclusion or exclusion of a short sequence of dark squares delimited by intersecting taglines. An error specific to the nnUnet pretrained with the physics-driven simulated tagged images is the presence of holes within the region of interest.

The ResNetVAE outperforms the nnUnet by 4.3 percentage points of average DSC and 1.10 mm of HD-95 (Table 3). Additionally, the training performance metrics are closer to the associated test scores for the ResNetVAE architecture. The nnUnet likely has too much capacity, leading to a tendency to overfit more in comparison to the smaller ResNetVAE with its regularized latent space.

#### 3.2.2. Training Strategies

The models that are pretrained perform better than the models that are not. It can be observed in Table 3 that pretrained models score better on both performance metrics. The average Dice score on the held-out test set is 2.7 and 1.4 percentage points higher when pretrained with simulated tagged images in comparison to training from scratch for the ResNetVAE and nnUnet architectures, respectively. On the other hand, the average 95th-percentile Hausdorff distance on the held-out test set is 0.87 and 1.4 mm lower for the ResNetVAE and nnUnet architectures, respectively.

In addition, the performance distribution has a larger variance and is thus less robust for the models trained from scratch or pretrained with *cine* images when compared to those pretrained with simulated tagged images (Table 3).

Finally, while both simulated tagged pretraining methods seem to converge similarly, the model trained from scratch is slower to converge and stagnates at a lower performance level Figure 8. It can also be observed that the models pretrained with simulated tagged data converge slightly faster than those pretrained with *cine* images, and more so for the ResNetVAE architecture.

#### 3.2.3. Acquisition Time-Frame

When looking at model performance versus acquisition time-frame in Figure 9, both architectures perform similarly across time. The distribution of the Dice coefficient is higher for images from the first 15 to 30 percent of the time series and worsens as the tag lines fade during diastole. However, the distribution of the DSC performance of the segmentation ResNet VAE is long-tailed starting around the middle section of the time series. Three examples of the fading tag lines and associated drop in performance are displayed in Figure 10. It seems that fading tag lines do not induce a specific type of error, as most inconsistent segmentations are similar across time frames.

#### 3.2.4. Pathology-Wise Performance

It can be observed in Figure 11 that the performance distributions on images from healthy subjects and DMD patients are similar for both models.

#### 3.2.5. Shape-Aware Loss

The performance metric distribution difference between null and non-null values of γ, the shape-aware loss improves learning performance by up to 3.9 Dice coefficient percentage points and 2.25 Hausdorff distance pixels (Table 4). According to the cumulative rank, the optimal contribution weight of the shape-aware loss to the optimization loss function is γ=0.05.

## 4. Discussion

In this work, a myocardium segmentation network for tagged CMR images reaching a Dice coefficient of 0.828 was implemented. Notably, a CycleGAN-based data augmentation strategy transforms public annotated *cine* datasets into their tagged-appearing equivalents to mitigate a low annotated data regime.

### 4.1. Analysis of the *Cine* to Tagged Image Transformation Models

While seemingly producing inconsistent contrast and despite not deforming tag lines through acquisition time, the physics-driven model implemented in this work appears to give an advantage when used to pre-train a myocardium segmentation model. Future research could evaluate the trade-off between the time and effort needed to reach higher levels of sophistication in the augmentation technique versus the added segmentation performance, specifically with regard to the propagation of motion of the tag lines in time.

The style transfer models developed in this work highlight the important potential of domain transformation models in biomedical imaging tasks. Previously used to augment datasets by transforming images taken by one vendor or research center to another [21], CycleGAN is also able to transfer domain-specific patterns from vastly different imaging modalities. The flexibility of this implementation enables further research to create more cardiac MRI-specific data augmentation models. For example, the simulation of respiratory motion artifacts or poor ECG gating. Additionally, resource- and time-intensive annotations can be shared across modalities thanks to the cycle-consistency constraint. Indeed, future work could include the LV and RV classes during model training as they are commonly available in public datasets. Cycle-consistent style transfer does bring some disadvantages, however. First, this type of model is difficult to evaluate quantitatively due to the generative nature of the task. Above that, it is prone to hallucinations, which can present an important problem in certain biomedical imaging applications. The model transforming a low-SNR domain into a high-SNR one usually presents vastly worse performance. In this work, for example, the tagged-to-*cine* model $G_{B\to A}$ shows many hallucinations and is unable to produce physiologically consistent-appearing structures. Finally, the CycleGAN used in this work is set with common baseline parameters for training. Future work could look into possible performance improvements using a more fine-grained tuning strategy for these parameters.

### 4.2. Analysis of the DL-Based Segmentation Network

The segmentation models presented in this work set a baseline performance standard as there have not been any efforts to segment the myocardium in tagged CMR images to the best of our knowledge. Above that, our top-performing model is almost on par with state-of-the-art myocardium segmentation in *cine* images despite having worse contrast and less distinguishable structures. The winners of the M&Ms challenge present an average DSC of 0.839 for ED slices.

For the pretraining strategy, we observe that using images close to the domain of the final application improves performance. Additionally, transfer learning vastly improves convergence and thus training efficiency.

While more commonly used in medical image segmentation, the nnUnet architecture is outperformed by the smaller and more regularized ResNetVAE. This result is likely due to an excess in capacity leading to overfitting the nnUnet on the training dataset. Moreover, qualitative analysis shows that the ResNetVAE architecture produces more anatomically accurate results. A reasonable explanation is that the regularization of the latent space for that architecture better incorporates anatomical constraints upstream of the decoding step.

As tag lines fade during acquisition, the image contrast varies, making the cardiac structures more or less distinguishable from their surroundings. Our results show that all models perform best between time frames four to seven out of twenty-five. Improvements are needed if full-cycle motion tracking is to be achieved.

While the difference is relatively small, the trained segmentation models perform better on healthy subjects compared to DMD patients. These patients often have trouble holding their breath during acquisition and their myocardium both contracts and appears differently, which seems to make the segmentation task more difficult. Nevertheless, the models trained in this work seem to generalize across different cardiac diseases. This observation emphasizes the importance of the inclusion of multiple diseases and conditions in the datasets used for training in DL-based CMR applications.

### 4.3. Future Work

The dataset used for the final training in this work is comprised of images from children; either healthy subjects or DMD patients. Additional research should extend myocardium segmentation models in tagged CMR images for adult subjects and of more diverse cardiovascular conditions. Model training and image analysis, in general, should shift to multi-domain approaches (multiple conditions such as SNR variations, centers, vendors, etc.) to promote the widespread development and subsequent deployment of clinical DL-based tools.

## 5. Conclusions

The objective of this work was to develop a myocardial segmentation model for tagged cardiac MR images in the absence of a large database of annotated tagged images for training. We used a CycleGAN to transform existing *cine* datasets with annotations into their tagged-appearing equivalents. By using these images for pretraining in conjunction with a small set of real tagged images used for training, we optimized common segmentation neural network architectures reaching a state-of-the-art average Dice coefficient of 0.828 on a held-out test set.

## Figures and Tables

**Figure 1 bioengineering-10-00166-f001:**
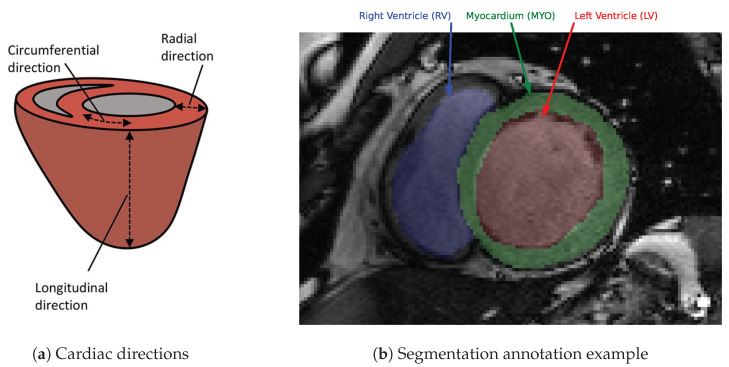
(**a**) Macro-scale diagram of the reference directions most commonly used in cardiac imaging, adapted from Wilson et al. (2022) [4]. (**b**) Annotated SAX CMR slice from the Automated Cardiac Diagnosis Challenge (ACDC) public dataset, MICCAI 2017 [5]. The expert annotations include segmentation masks for the LV blood pool, myocardium, and RV (respectively indicated in red, green, and blue). This particular subject was diagnosed with Dilated Cardiomyopathy (DCM).

**Figure 2 bioengineering-10-00166-f002:**
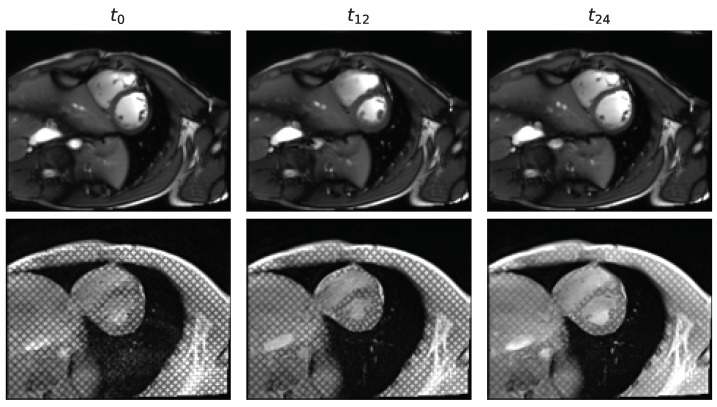
Example of SAX *cine* and associated tagged cardiac MR images through time from a control patient. Times {0,12,24} are displayed from recordings of 25 time-frames.

**Figure 3 bioengineering-10-00166-f003:**
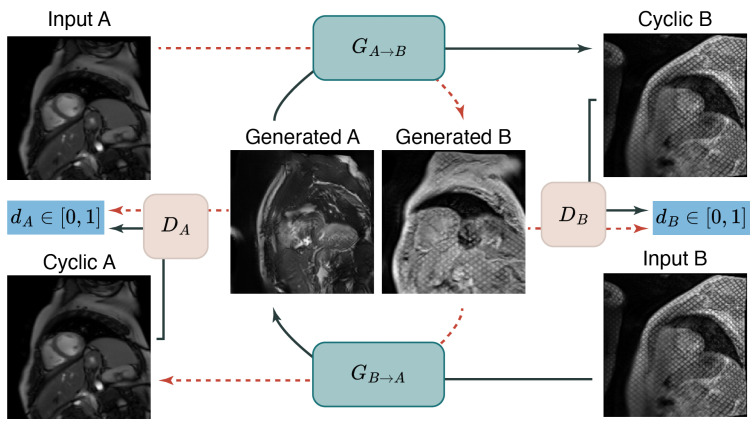
CycleGAN architecture. Training was performed using unlabeled images from groups A and B (*cine* and tagged MR images, respectively) for unpaired image-to-image translation. Each discriminator encourages the associated generator to transform images from one domain to outputs that are indistinguishable from the other domain. Additionally, training includes a pixel-wise cycle consistency constraint that minimizes the dissimilarity between arbitrary input images from one domain and its cyclic output.

**Figure 4 bioengineering-10-00166-f004:**
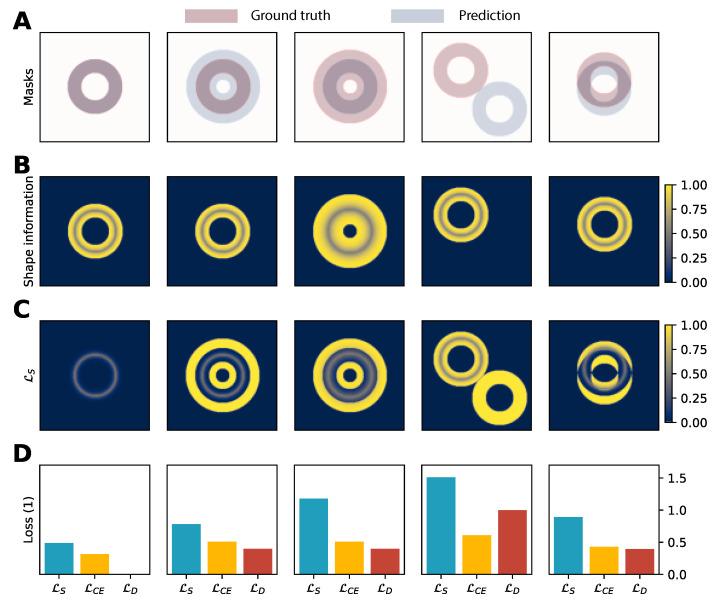
Toy example comparison of the shape-aware loss ($L_{S}$; Equation (Equation 3)) to cross-entropy loss ($L_{CE}$) and Dice loss ($L_{D}$), specifically discriminating on the boundary of the ground truth mask. (**A**) Ground truth and prediction masks were used as toy examples. (**B**) Shape information SI of each ground truth mask. (**C**) shape-aware loss $L_{S}$ before channel- and example-wise aggregation as defined in Equation (Equation 3). (**D**) Bar-chart of the different losses.

**Figure 5 bioengineering-10-00166-f005:**
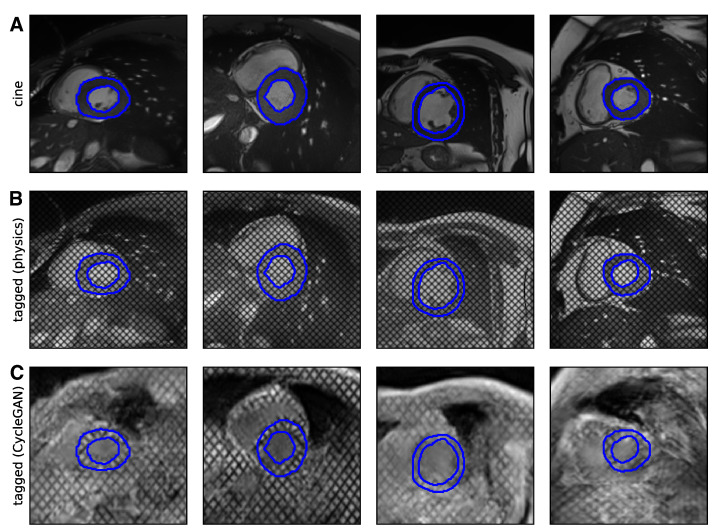
Examples of *cine* to tagged image transformation performed by (**B**) the physics-driven transformation and (**C**) the trained CycleGAN on (**A**) a set of labeled images from the ACDC dataset, completely disjoint from the dataset used to train the CycleGAN.

**Figure 6 bioengineering-10-00166-f006:**
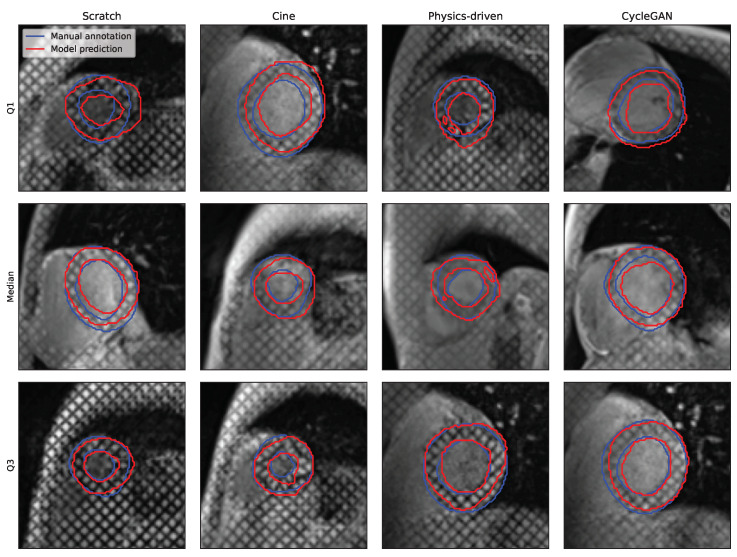
Qualitative segmentation results for the nnUnet model architecture and different pretraining strategies (with cine images, simulated tagged with physics-driven transformation, and simulated tagged with CycleGAN transformation). The Q1, median, and Q3 examples are chosen from each results distribution based on DSC.

**Figure 7 bioengineering-10-00166-f007:**
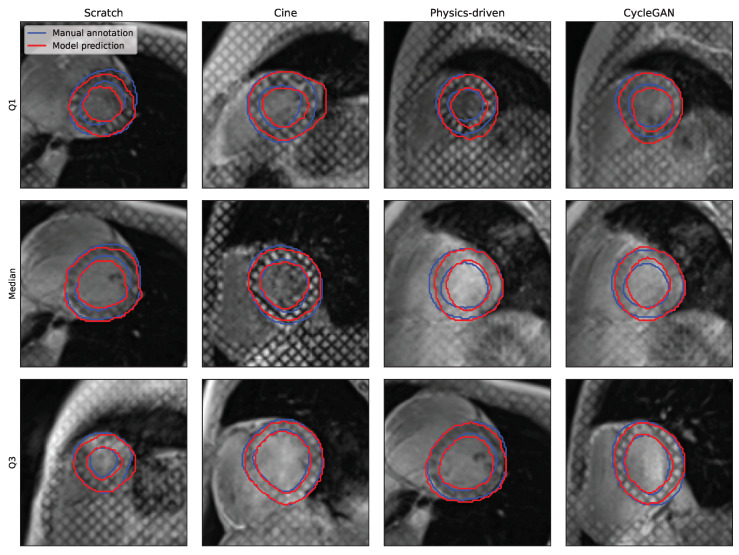
Qualitative segmentation results for the ResNetVAE model architecture and different pretraining strategies (with cine images, simulated tagged with physics-driven transformation, and simulated tagged with CycleGAN transformation). The Q1, median, and Q3 examples are chosen from each results distribution based on DSC.

**Figure 8 bioengineering-10-00166-f008:**
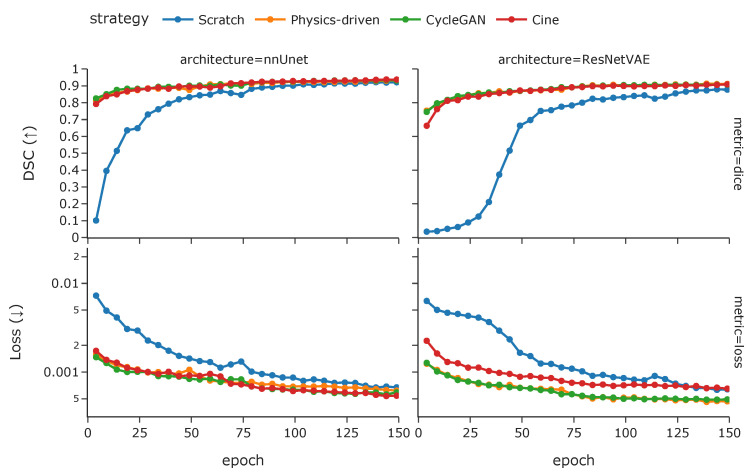
The training Dice coefficient and loss during the 150-epoch weight optimization procedure for different model architectures and pretraining strategies (with cine images, simulated tagged with physics-driven transformation, and simulated tagged with CycleGAN transformation).

**Figure 9 bioengineering-10-00166-f009:**
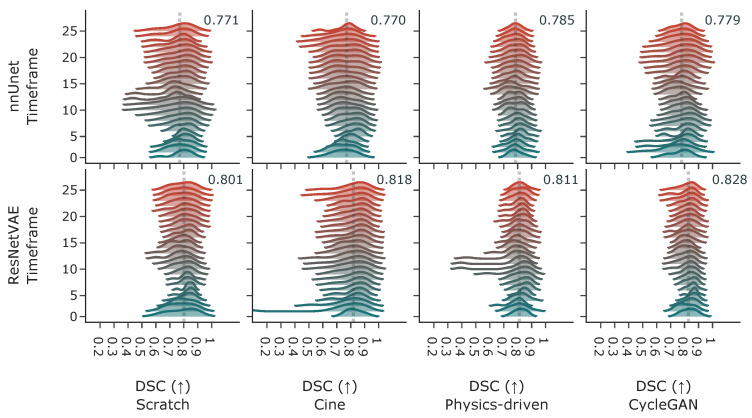
Dice coefficient performance metric distribution over the acquisition time-frames for nnUnet and ResNetVAE model architectures and different pretraining strategies (with cine images, simulated tagged with physics-driven transformation, and simulated tagged with CycleGAN transformation). The annotated dotted vertical line displays the average dice coefficient in each subplot. Time-point 1 is at ED and time-point 25 is at ES.

**Figure 10 bioengineering-10-00166-f010:**
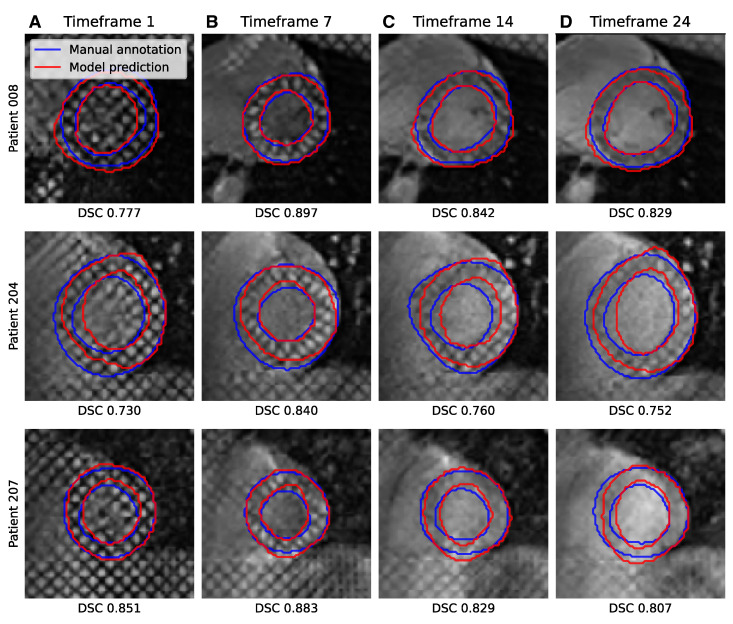
Example segmentations of acquired tagged CMR images from three patients from the test set for image frames 1, 7, 14, and 24 (respectively displayed in columns **A**–**D**) and associated DSC performance scores. The model predictions are generated by the neural network with the ResNetVAE architecture and CycleGAN pretraining strategy.

**Figure 11 bioengineering-10-00166-f011:**
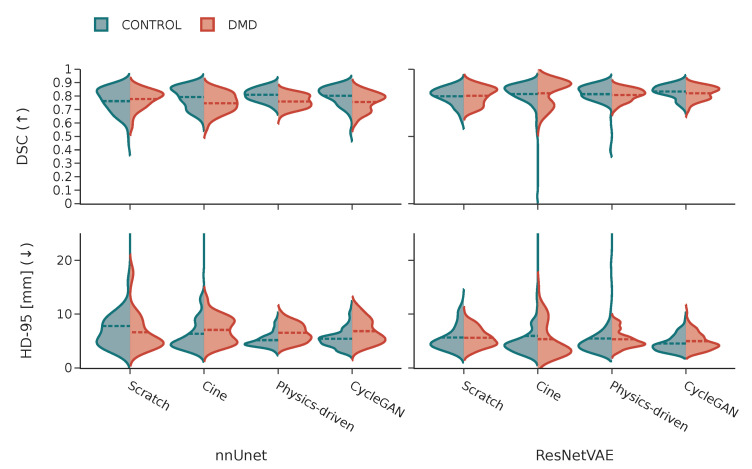
Comparative violin plots displaying the difference in performance metric distributions between control and DMD patients for each model for different model architectures and different pretraining strategies (with cine images, simulated tagged with physics-driven transformation, and simulated tagged with CycleGAN transformation).

**Table 1 bioengineering-10-00166-t001:** Description of publicly available *cine* datasets used in this work, all published as workshop challenges for MICCAI. The number of images consists of 2D slices available for training. All datasets contain healthy subjects. The possible annotations are segmentation masks of the LV, myocardium (MYO), and RV.

Name	Date	Subjects	Pathologies	Centres	Vendors	Images	Labels
ACDC [5]	2017	150	4	1	1	1828	LV, RV, MYO
M&Ms [10]	2020	375	9	6	4	2468	LV, RV, MYO
SCD [15]	2009	15	3	1	1	282	LV, MYO

**Table 2 bioengineering-10-00166-t002:** Training and inference statistics for the trained network architectures. Training is performed on batches of 128 images of size [256, 256] in parallel on two TITAN RTX GPUs with 24 GB of memory each. Inference time statistics are measured on seven runs of a batch of one image.

Architecture	No. Parameters	Training Time	Inference Time
**ResNetVAE**	3555891	64.3 s/epoch	112 ms ± 28.7 ms
**nnUnet**	34164258	163.2 s/epoch	661 ms ± 274 ms

**Table 3 bioengineering-10-00166-t003:** Performance metrics for the evaluated architectures, training strategies, and data splits. Displayed are the Dice coefficient (DSC) and 95th percentile Hausdorff Distance (HD-95) for which higher and lower are better, respectively. The average (Avg.) and standard deviation (Std.) statistics are computed over the training and testing data splits containing 1000 and 250 2D SAX tagged MR images, respectively.

		DSC (↑)				HD-95 [mm] (↓)			
	Split	Train		Test		Train		Test	
Model	Strategy	Avg.	Std.	Avg.	Std.	Avg.	Std.	Avg.	Std.
**nnUnet**	**Scratch**	0.894	0.021	0.771	0.084	2.519	0.554	7.219	5.075
	**Cine**	0.895	0.020	0.770	0.078	2.325	0.545	6.700	3.435
	**Physics-driven**	0.895	0.020	**0.785**	0.054	3.115	0.871	**5.846**	1.625
	**CycleGAN**	0.894	0.020	0.779	0.074	2.385	0.569	6.117	2.087
**ResNetVAE**	**Scratch**	0.872	0.027	0.801	0.065	3.301	0.746	5.616	1.989
	**Cine**	0.820	0.100	0.818	0.096	6.621	8.741	5.647	5.549
	**Physics-driven**	0.895	0.023	0.811	0.068	2.517	0.564	5.407	3.350
	**CycleGAN**	0.898	0.022	**0.828**	0.049	2.481	0.572	**4.745**	1.537

**Table 4 bioengineering-10-00166-t004:** Performance metrics for the shape distance loss weight factor γ. Displayed are the Dice coefficient (DSC) and 95th percentile Hausdorff Distance (HD-95) for which higher and lower are better, respectively. Each model is a ResNetVAE architecture trained from scratch on tagged CMR images for 200 epochs. The average (Avg.) and standard deviation (Std.) statistics are computed over the training and testing dataset splits containing 1000 and 250 2D SAX-tagged MR images, respectively. The rank is based on the cumulative ranking for both evaluation metrics.

	DSC (↑)				HD-95 [mm] (↓)				Rank
Split	Train		Test		Train		Test		
γ	Avg.	Std.	Avg.	Std.	Avg.	Std.	Avg.	Std.	
0.000	0.880	0.025	0.768	0.094	3.082	0.668	7.701	8.967	7
0.001	0.815	0.025	0.771	0.045	3.375	0.771	**5.358**	5.946	2
0.005	0.886	0.023	0.770	0.102	2.976	0.658	7.628	9.414	4
0.050	0.872	0.027	**0.801**	0.065	3.301	0.746	5.616	1.989	1
0.100	0.865	0.026	0.798	0.083	3.547	0.849	6.423	5.314	3
0.500	0.789	0.023	0.736	0.050	3.496	0.958	6.449	10.825	6
1.000	0.585	0.035	0.556	0.041	4.838	3.672	5.715	1.975	4

## Data Availability

The data is available with the corresponding author. A part or all the data can be shared upon reasonable request.

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
