# Peer review of "Myocardial Segmentation of Tagged Magnetic Resonance Images with Transfer Learning Using Generative Cine-To-Tagged Dataset Transformation"

_bioengineering, 2023, doi:10.3390/bioengineering10020166_

Round 1

Reviewer 1 Report

In this paper, two deep learning models for myocardium segmentation from tagged CMR images are developed and compared, which can well exploit structural information from large databases of unannotated cine images. Based on the pre-trained model, the myocardial segmentation network achieves a state-of-the-art Dice coefficient of 0.828 and converges faster. This paper is well organized generally but there are some concerns as follows.

Q1: This paper lacks a detailed description of the model structure. Except for the comparative overview information provided in Table 2, we cannot know the details about the network architecture, such as the number of network layers and activation functions, and the details of training are not clear, such as the setting of the learning rate, training strategies, etc.

Q2: The notation of some formulas in the text needs to be clarified, which is confusing. For example, the notation "C" in Equation (3) and (4) need to be further explained its specific meaning. The k is set to 0.2 in Equation (1) and the author should explain why this value is chosen and whether more experiments should be conducted on this super parameter.

Q3:In the experimental part, the author provides the loss and Dice change curves during the training and verification process, reflecting the convergence of the two models. However, each subgraph in Figure 8 contains 8 different parameter settings, and the contents are displayed in a mess, even some are indistinguishable. It is recommended that the author separate multiple subgraphs at different stages and plot the curves using easily distinguishable colors.

Q4: The image transformation of the proposed physics-driven method relies on empirical selection, and there is a lack of ablation contrast experiments to verify its effectiveness. CycleGAN is a relatively mature method in the computer vision field, but the author has not made major improvements, and the experiments in this paper lack comparisons of different network parameters. The author needs to provide more detailed training strategy information to explain how the physical drive method can achieve better performance.

Q5: Medical image computing (including segmentation, super-resolution, augmentation

etc.) using GAN based model is a hot topic today. Authors should give a more extensive review in this topic. There are some representative works, Such as:

Fine perceptive GANs for brain MR image super-resolution in wavelet domainï¼›

Bidirectional Mapping Generative Adversarial Networks for Brain MR to PET Synthesis;

Brain Stroke Lesion Segmentation Using Consistent Perception Generative Adversarial Network.

Q6:There are some typos and grammatical errors in the text. For example, these words "best performing", "5 pixel" and "long tailed" all miss a hyphen. The word "automization" in line 36 should be corrected as "automation". The verb "is" in the sentence "This is sometimes appears exaggerated..." is unnecessary; "are" in the sentence "While the difference are relatively small" should be "is". These mistakes should be rechecked and corrected carefully to avoid similar problems in this paper.

Author Response

Response to Reviewer 1 Comments

Point 1: This paper lacks a detailed description of the model structure. Except for the comparative overview information provided in Table 2, we cannot know the details about the network architecture, such as the number of network layers and activation functions, and the details of training are not clear, such as the setting of the learning rate, training strategies, etc.

Response to point 1: Details about the network architecture have been added in Section 2.2.3. We have further addressed the reviewer’s point by adding more fine-grained training details to Section 2.2.5.

Point 2: The notation of some formulas in the text needs to be clarified, which is confusing. For example, the notation "C" in Equation (3) and (4) need to be further explained its specific meaning. The k is set to 0.2 in Equation (1) and the author should explain why this value is chosen and whether more experiments should be conducted on this super parameter.

Response to point 2: The definition of the variable C, i.e. the number of image channels, used in Equations (3) and (4) has been updated. An amendment to Section 2.2.5 further explains that the value of k was set to 0.2 by experimental selection.

Point 3: In the experimental part, the author provides the loss and Dice change curves during the training and verification process, reflecting the convergence of the two models. However, each subgraph in Figure 8 contains 8 different parameter settings, and the contents are displayed in a mess, even some are indistinguishable. It is recommended that the author separate multiple subgraphs at different stages and plot the curves using easily distinguishable colors.

Response to point 3: We agree with the reviewer that the formatting of Figure 8 can be improved. To do so, we’ve (1) removed the dotted validation performance curves, (2) added scatter-point markers, and (3) modified the colors to make them more distinguishable. The objective is to make the comparison of the performance of the different pretraining strategies more visible. As such, the main observation is that the models that are pre-trained converge faster and reach higher performance than those that are not. As an additional note: while the validation curves have been removed, the training and testing performance metrics remain available in Tables 3 and 4 to evaluate potential over-fitting. 

Point 4: The image transformation of the proposed physics-driven method relies on empirical selection, and there is a lack of ablation contrast experiments to verify its effectiveness. CycleGAN is a relatively mature method in the computer vision field, but the author has not made major improvements, and the experiments in this paper lack comparisons of different network parameters. The author needs to provide more detailed training strategy information to explain how the physics-driven method can achieve better performance.

Response to point 4: The parameters used in the physics-driven method were based on contrast mechanisms that have been extensively tested and selected in Loecher et al. (2020). As its training is independent of gradient optimization techniques, Section 2.1.2 was amended with additional information relevant to the computational algorithm developed.

Regarding the generative method, we agree that our work didn't contain an in-depth sensitivity analysis of the model hyperparameters. We used the commonly accepted baseline values for this method that have been well validated in other works. Indeed, now that this study has established that pretraining on style-transfer tagged images provides a significant training advantage, we may seek to further improve the CycleGAN architecture and hyperparameters in future work. To address this point, the discussion of the manuscript was amended accordingly.

Point 5: Medical image computing (including segmentation, super-resolution, augmentation etc.) using GAN based models is a hot topic today. Authors should give a more extensive review in this topic. There are some representative works, Such as:

  • Fine perceptive GANs for brain MR image super-resolution in wavelet domainï¼›
  • Bidirectional Mapping Generative Adversarial Networks for Brain MR to PET Synthesis;
  • Brain Stroke Lesion Segmentation Using Consistent Perception Generative Adversarial Network

Response to point 5: To address the reviewer’s point, an amendment has been made to the introduction (Section 1) to give additional context regarding the topic of the use of GAN based models in medical imaging.

Point 6: There are some typos and grammatical errors in the text. For example, these words "best performing", "5 pixel" and "long tailed" all miss a hyphen. The word "automization" in line 36 should be corrected as "automation". The verb "is" in the sentence "This is sometimes appears exaggerated..." is unnecessary; "are" in the sentence "While the difference are relatively small" should be "is". These mistakes should be rechecked and corrected carefully to avoid similar problems in this paper.

Response to point 6: The pointed out typos and grammatical errors have been fixed in the revised manuscript.

Reviewer 2 Report

Dear Authors,

This proposed work is presented using the clinical data and the methodology and the results are fine.

I request you to consider the following suggestions:

1. Please discuss the necessary initial image processing methodology considered to treat the clinical MRI (if any procedure implemented). Also confirm the MRI is in 3D for or 2D?  This may help the user to understand the necessary adjustment to be implement on clinical data, when is to be analyzed with a computer algorithm

2. The number of parameters in Table 2 is confusing; If possible, please change (Ex. 3’555’891 to 3555891)

3. Data Availability Statement, need to be modified:

Ex. (i) Data cannot be shared due to ethical issues (or)

(ii) The data is available with the corresponding author. A part or all the data can be shared upon request

Author Response

Point 1: Please discuss the necessary initial image processing methodology considered to treat the clinical MRI (if any procedure implemented). Also confirm the MRI is in 3D for or 2D?  This may help the user to understand the necessary adjustment to be implement on clinical data, when is to be analyzed with a computer algorithm

Response to point 1: We confirm that the MRI images are 2D and have clarified this in Section 2.2.1. Additionally, the initial image processing methodology is encapsulated in the preprocessing pipeline and consists of two steps: (1) resizing the image to 256 x 256 (not necessary for the nnUnet) and (2) normalization of the input image using a mean of 0.456 and a standard deviation of 0.224. These details have been clarified and amended to Section 2.2.3.

Point 2: The number of parameters in Table 2 is confusing; If possible, please change (Ex. 3’555’891 to 3555891)

Response to point 2: The mentioned stylistic number representation modification was performed.

Point 3: Data Availability Statement, need to be modified: Ex. (i) Data cannot be shared due to ethical issues (or) (ii) The data is available with the corresponding author. A part or all the data can be shared upon request.

Response to point 3: The data availability statement was modified as requested to the second suggested option.

Round 2

Reviewer 1 Report

My previous concerns have been addressed.